# Large-scale outbreak of Chikungunya virus infection in Thailand, 2018–2019

**Sarawut Khongwichit**[1], **Jira Chansaenroj**[1], **Thanunrat Thongmee**[1],
**Saovanee Benjamanukul**[ORCID][2], **Nasamon Wanlapakorn**[1,3], **Chintana Chirathaworn**[4,5]*,
**Yong Poovorawan**[ORCID][1]*

**1** Department of Pediatrics, Center of Excellence in Clinical Virology, Faculty of Medicine, Chulalongkorn University, Bangkok, Thailand, **2** Department of Medicine, Banphaeo General Hospital, Samutsakhon, Thailand, **3** Division of Academic Affairs, Faculty of Medicine, Chulalongkorn University, Bangkok, Thailand, **4** Department of Microbiology, Faculty of Medicine, Chulalongkorn University, Bangkok, Thailand, **5** Tropical Medicine Cluster, Chulalongkorn University, Bangkok, Thailand

\* yong.p@chula.ac.th (YP); chintana.ch@chula.ac.th (CC)

**Data Availability Statement:** All relevant data are within the manuscript and its Supporting Information files.

**Funding:** This work was supported by the Research Chair Grant from the National Science

## Abstract

Between 2018 and 2019, the incidence of chikungunya was approximately 15,000 cases across 60 provinces in Thailand. Here, the clinical presentations in chikungunya, emergent pattern, and genomic diversity of the chikungunya virus (CHIKV) causing this massive outbreak were demonstrated. A total of 1,806 sera samples from suspected cases of chikungunya were collected from 13 provinces in Thailand, and samples were tested for the presence of CHIKV RNA, IgG, and IgM using real-time PCR, enzyme-linked immunoassay (ELISA), commercial immunoassay (rapid test). The phylogenetic tree of CHIKV whole-genome and CHIKV E1 were constructed using the maximum-likelihood method. CHIKV infection was confirmed in 547 (42.2%) male and 748 (57.8%) female patients by positive real-time PCR results and/or CHIKV IgM antibody titers. Unsurprisingly, CHIKV RNA was detected in >80% of confirmed cases between 1 and 5 days after symptom onset, whereas anti-CHIKV IgM was detectable in >90% of cases after day 6. Older age was clearly one of the risk factors for the development of arthralgia in infected patients. Although phylogenetic analysis revealed that the present CHIKV Thailand strain of 2018–2020 belongs to the East, Central, and Southern African (ECSA) genotype similar to the CHIKV strains that caused outbreaks during 2008–2009 and 2013, all present CHIKV Thailand strains were clustered within the recent CHIKV strain that caused an outbreak in South Asia. Interestingly, all present CHIKV Thailand strains possess two mutations, E1-K211E, and E2-V264A, in the background of E1-226A. These mutations are reported to be associated with virus-adapted *Aedes aegypti*. Taken together, it was likely that the present CHIKV outbreak in Thailand occurred as a result of the importation of the CHIKV strain from South Asia. Understanding with viral genetic diversity is essential for epidemiological study and may contribute to better disease management and preventive measures.

and Technology Development Agency (P-15-50004), the Center of Excellence in Clinical Virology of Chulalongkorn University and Hospital. SK is supported by Second Century Fund (C2F), Chulalongkorn University. The funding agencies had no role in the study design, data collection and analysis, decision to publish, or preparation of the manuscript.

**Competing interests:** NO authors have competing interests.

## Introduction

Chikungunya is a vector-borne infectious disease caused by the Chikungunya virus (CHIKV), an *Alphavirus* in the family *Togaviridae*. CHIKV infection was initially identified in 1952 in a febrile patient in Tanzania, Africa [1]. CHIKV can be transmitted to humans by *Aedes aegypti* and *Aedes albopictus* mosquitoes. The disease is characterized by fever, skin rash, and joint pain or arthralgia. The common consequences of CHIKV infection are severe arthritis and polyarthralgia, which can persist for weeks or months. However, neurological problems including encephalitis, myelopathy, peripheral neuropathy, myeloneuropathy, and myopathy, have also been reported [2, 3]. There is currently no effective antiviral drug or vaccine to treat or prevent CHIKV infection, respectively. Real-time reverse transcription-polymerase chain reaction (RT-PCR) for the detection of CHIKV RNA and testing of IgM antibody specific to CHIKV antigen are widely used for laboratory diagnosis of CHIKV infection [4–6]. To date, CHIKV infection has been detected in over 100 countries and millions of people worldwide have been infected.

CHIKV is an enveloped RNA virus. The non-structural proteins (nsP), capsid (C), and envelope (E) proteins are encoded by the 11.8-kb CHIKV genome [7, 8]. Nucleotide and amino acid sequence studies from different isolates of CHIKV have shown that there are three major lineages of CHIKV–East/Central/South African (ECSA), West African (WA), and Asian [9, 10].

The first CHIKV outbreak in Thailand was reported in Bangkok in 1958. The CHIKV Asian genotype was the cause of this outbreak [11]. Another outbreak in Thailand occurred in 2008–2009 and affected mainly the southern region of the country. The ECSA genotype with the mutation of amino acid residue 226 from alanine to valine in the E1 envelope glycoprotein (E1-A226V) was reported [12]. The same genotype with E1-A226V mutation was detected in the CHIKV outbreak in northeast Thailand in 2013 as a result of CHIKV circulating within the country [13]. The number of cases reported was lower in 2010 and continuously declined until several years later, when the large outbreak started in 2018–2019. CHIKV spread nation-wide and approximately 15,000 confirmed cases were reported between 2018 and 2019. According to the Bureau of Epidemiology, Ministry of Public Health, Thailand, the number of monthly reported chikungunya cases started to rise in June 2018. All of the initial cases were individuals residing in Satun and Narathiwas provinces in the south of Thailand. The number of reported cases continuously rose from <20 cases per month between January and May to 1,171 and 1,759 cases in November and December 2018, respectively [14]. In 2019, the outbreak was reported in 60 provinces across the country but the majority of cases were from Bangkok and several provinces in the south of Thailand. Overall, the morbidity rates of chikungunya in Thailand in 2018 and 2019 were 5.40 and 19.73 per 100,000 population, respectively.

We previously reported the genome sequences of CHIKV isolates from the outbreak in Satun and southwest Bangkok during the 2018 monsoon. The causative CHIKV strains belong to the ECSA genotype; however, these ECSA strains did not carry E1-A226V mutation [15]. This study focused on the clinical presentations, emergent patterns, and genomic diversity of CHIKV that caused a large-scale outbreak across Thailand between October 2018 and February 2020. Samples were collected from various parts of the country during the large-scale outbreak. Molecular characterization of CHIKV isolates, including the whole genome and E1 gene sequencing was performed. In addition, laboratory diagnosis by real-time PCR, ELISA IgM/IgG, and commercial fluorescence immunoassay IgM/IgG (rapid test) were compared. The large pool of samples with well-recorded symptoms and the number of days after

symptom onset were analyzed along with the laboratory results, which demonstrated the importance of proper diagnostic tests to confirm CHIKV infection.

## Materials and methods

The research proposal was approved by the institutional review board of the Ethics Committee of the Faculty of Medicine, Chulalongkorn University, Thailand (IRB no. 630/61). The IRB waived the need for informed consent from the participants, because the clinical specimens were anonymous. The study adhered to the Declaration of Helsinki and Good Clinical Practice Guidelines (ICH-GCP).

### Sample collection

Serum or plasma samples were collected from patients with suspected chikungunya along with demographic and clinical data (age, sex, fever, joint pain, rash, and conjunctivitis). A suspected case of CHIKV was defined as a patient presenting with acute onset of fever ($> 38.5°C$) with or without severe joint pain and skin rash, particularly in a person who residing, traveling, and working in an epidemic area with a high risk of CHIKV transmission. Samples were sent for laboratory diagnosis at the Center of Excellence in Clinical Virology, Faculty of Medicine, Chulalongkorn University, Bangkok, Thailand. From October 2018 through February 2020, 1806 samples were collected from 13 provinces throughout the Central (Bangkok 1,137 cases, Samut Prakan 173 cases, Samut Sakhon 344 cases, Nonthaburi 1 case); Western (Ratchaburi 1 case, Kanchanaburi 1 case); Northern (Tak 1 case); Northeastern (Maha Sarakham 1 case); Eastern (Chon Buri 34 cases); and Southern (Ranong 1 case, Phuket 108 cases, Songkhla 1 case, Surat Thani 3 cases) regions of Thailand (Fig 1). A confirmed case of chikungunya was defined as a suspected case plus laboratory confirmation either by real-time RT-PCR to detect CHIKV viral RNA or serological test to detect CHIKV-specific IgM in serum. Patients possessing CHIKV-specific IgG without IgM were classified as having had a past CHIKV infection.

### Detection of CHIKV by real-time reverse transcription-polymerase chain reaction (RT-PCR)

Total RNA from serum samples was extracted using magLEAD® kit and amplified by qualitative real-time RT-PCR based on fluorescent hybridization probe assay using the primer sets and PCR conditions as described in a previous report (1) Detection was performed using LightCycler® 480 Instrument II (Roche, Basel, Switzerland).

### Detection of anti-CHIKV IgM and IgG antibodies by rapid test

In this study, IgM/IgG antibodies against CHIKV in all serum samples were assessed using commercial fluorescence immunoassay on nitrocellulose membranes of a test device (SD BIO-SENSOR, Gyeonggi-do, Korea) according to the manufacturer's instruction. Briefly, serum was added to the membrane of the test device, which consisted of immobilized anti-human IgM or anti-human IgG antibodies on two individual test lines. The assay diluent was then added to release the chikungunya antigen from the antigen pad, and the anti-CHIKV envelope monoclonal antibody conjugated with europium from the conjugation pad. The intensity of the fluorescence signal was evaluated with a STANDARD F200 Analyzer (SD BIOSENSOR, Gyeonggi-do, Korea).

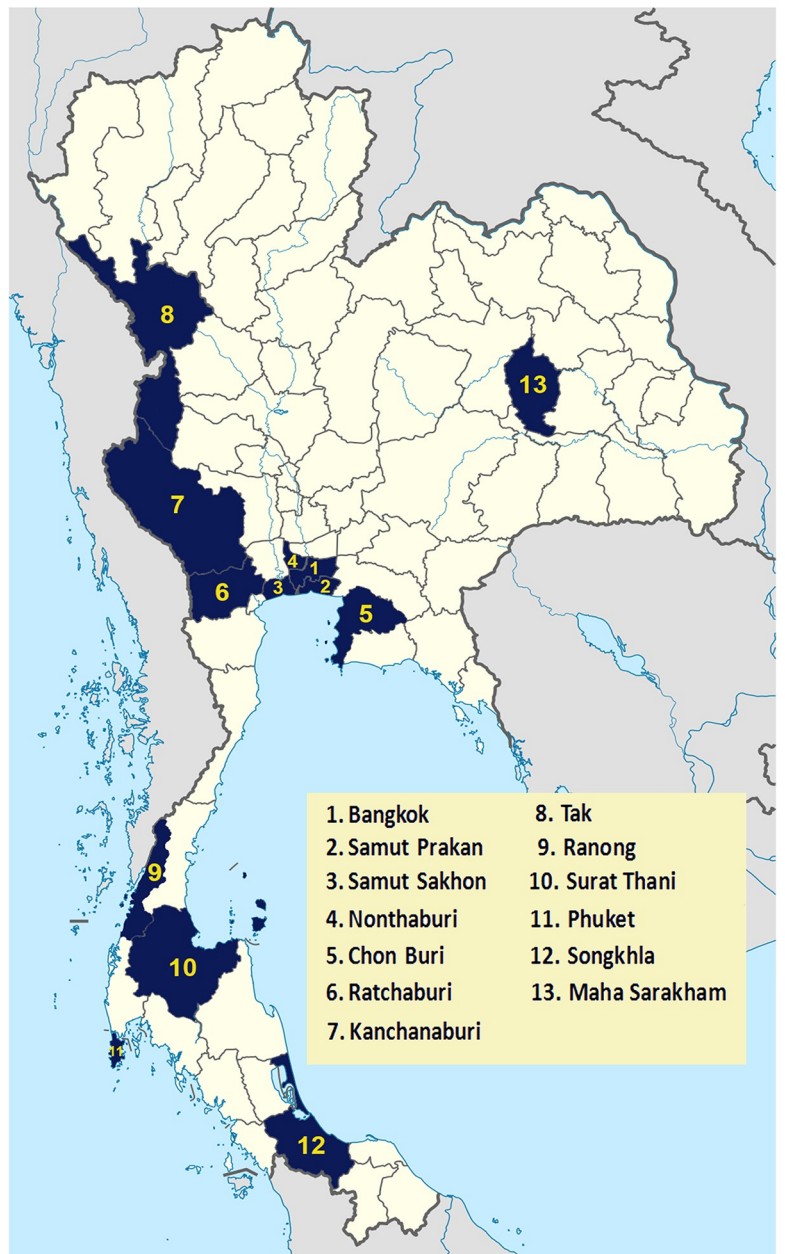

**Fig 1. Map of Thailand showing the sample collection site in the 13 provinces.** Map of Thailand showing sampling areas in Bangkok (1,137 cases), Samut Prakan (173 cases), Samut Sakhon (344 cases), Nonthaburi (1 case), Ratchaburi (1 case), Kanchanaburi (1 case), Tak (1 case), Maha Sarakham (1 case), Chon Buri (34 cases), Ranong (1 case), Phuket (108 cases), Songkhla (1 case), and Surat Thani (3 cases).

## Detection of anti-CHIKV IgM and IgG antibodies by Enzyme-Linked Immunosorbent Assay (ELISA)

In this study, the commercial ELISA kits (EUROIMMUN, Lübeck, Germany) [16] were performed to determine anti-CHIKV IgM/IgG antibodies with only serum from the first consecutive 310 CHIKV-infected patients whose sera samples were positive for CHIKV RNA by real-time PCR or positive for anti-CHIKV IgM by rapid test. The assay was performed as

recommended by the manufacturer. In brief, patient serum (diluted 1:100) was added to the microplate coated with the recombinant CHIKV protein. After incubation, the wells were washed and incubated with peroxidase-conjugated anti-human IgM/IgG. After removal of the unbound antibodies, the substrate solution was added, and the colorimetric output was measured at 450 nm after the enzymatic reaction was stopped. The OD ratios obtained from each sample were interpreted according to the manufacturer's recommendations.

### Genotypic characterization of CHIKV

CHIKV whole-genome and envelope 1 (E1) gene were amplified from the positive real-time RT-PCR samples by using primer sequences as previously reported [17]. The amplified products were excised and purified with the Expin™ Combo GP kit (General Biosystem, South Korea) as per the manufacturer's instructions. The purified PCR products were subjected to sequencing analysis (1st BASE, Singapore). The nucleotide sequences were examined using NCBI BLAST program (http://blast.ncbi.nlm.nih.gov/Blast.cgi) and were edited using CHROMAS LITE v. 2.0 program. The edited sequences were assembled by using BioEdit Sequence Alignment Editor v. 7.0.5.3 program. The obtained sequences were aligned with the known CHIKV sequences representing various genotypes from GenBank. The phylogenetic trees of CHIKV whole-genome and E1 gene were constructed using MEGA X program. All sequences were deposited in the GenBank database. The GenBank accession numbers of each sequence are shown in S1 File.

### Data analysis

The epidemiological surveillance of patients with CHIKV infection was performed by the Bureau of Epidemiology, MoPH, Thailand, and our recorded data. We retrieved these data, tabulated, and analyzed them using GraphPad Prism8 (GraphPad Software Inc. San Diego, CA) and Microsoft Excel (Microsoft Inc., Redmond, WA). The qualitative data were shown as n (%). The correlation analysis between the date of onset and the presence of IgM and CHIKV RNA was plotted and fit by simple linear regression with correlation coefficient $R^2$. The agreement between ELISA assay and rapid test were assessed by Cohen's kappa value. A chi-square test and logistic regression analysis were performed to examine the association between patient demographics including sex, age, and joint pain. Statistical analysis of significance was analyzed by using SPSS software, version 22. *P* value of less than 0.05 was considered as statistically significant.

## Results

### Chikungunya virus infection in Thailand

We investigated samples from 1,806 patients with suspected CHIKV infections that were sent to our center from October 2018 to February 2020. Among 1,806 suspected cases, 1,295 (547 male and 748 female) patients were confirmed to have CHIKV infection based on positive results of real-time RT-PCR and/or IgM antibody testing. As shown in Fig 2, the peak of CHIKV-positive cases in 2018 was reached in December, which was similar to the report by MoPH, Thailand. As mentioned above, a large-scale outbreak in Thailand was observed in 2019. The highest number of positive cases detected in our center was 372 cases in November 2019.

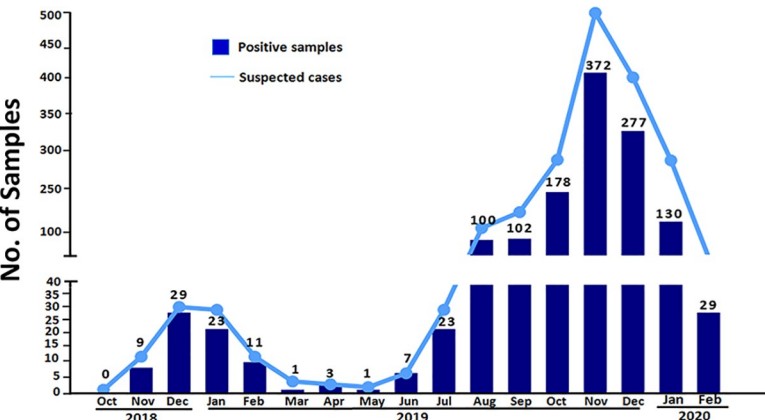

**Fig 2. Suspected and confirmed chikungunya cases between October 2018 and February 2020.** The number of suspected and confirmed cases whose samples were sent to the Center of Excellence in Clinical Virology, Faculty of Medicine, Chulalongkorn University. The numbers on bar graphs indicated the numbers of positive cases detected in each month between October 2018 and February 2020.

### Demographic and clinical presentation

CHIKV-infected patients were further classified according to age groups and symptoms (Table 1). The age ranges of patients, including in this study, were 3 to 97 years old. We found that chikungunya fever occurred in all ages. According to this study, the percentages of CHIKV-confirmed cases was higher in females than males (57.8 vs. 42.2%). The presence of joint pain was slightly lower in males (70.2%) than in females (74.6%), but these differences did not reach statistical significance (p = 0.079 for sex and p = 0.089 for the presence of joint pain). The most affected age range was between 31 and 40 years, with 299 cases (23.1%). Patients infected with CHIKV commonly presented with fever (96.6%), rash (50.2%), and

**Table 1. CHIKV-infected patients classified according to the presence of joint pain and age groups.**

|  | Number of patients (%) | Number of patients with joint pain (%) | p value[a] | OR (95% CI)[b] | p value[b] |
|---|---|---|---|---|---|
| Sex |  |  |  |  |  |
| Male | 547 (42.2%) | 384 (70.2%) | 0.079 | Reference | Reference |
| Female | 748 (57.8%) | 558 (74.6%) |  | 0.8 (0.6–1.0) | 0.089 |
| Age (years) |  |  | <0.001 |  |  |
| <1–10 | 92 (7.10%) | 33 (35.9%) |  | 0.3 (0.2–0.5) | <0.001 |
| 11–20 | 130 (10.0%) | 87 (66.9%) |  | Reference | Reference |
| 21–30 | 201 (15.5%) | 150 (74.6%) |  | 1.5 (0.9–2.4) | 0.130 |
| 31–40 | 299 (23.1%) | 223 (74.6%) |  | 1.5 (0.9–2.3) | 0.104 |
| 41–50 | 267 (20.6%) | 205 (76.8%) |  | 1.6 (1.0–2.6) | 0.380 |
| 51–60 | 187 (14.4%) | 147 (78.6%) |  | 1.8 (1.1–3.0) | 0.021 |
| >61 | 119 (9.2%) | 97 (81.5%) |  | 2.2 (1.2–3.9) | 0.010 |
| Symptom |  |  |  |  |  |
| Fever | 1251 (96.6%) |  |  |  |  |
| Arthralgia | 942 (72.7%) |  |  |  |  |
| Rash | 650 (50.2%) |  |  |  |  |

[a] Chi-square test

[b] Logistic Regression analysis

joint pain (72.7%). Joint pain is a common presenting symptom similarly observed in patients with rheumatoid arthritis and is a significant cause of distress in patients with chikungunya. In our study, 72.7% (942/1295) patients complained of joint pain. CHIKV-infected patients with joint pain were further analyzed according to age groups. As shown in Table 1, the presence of joint pain in CHIKV-confirmed cases was associated with age ($p < 0.001$). The highest percentage (81.5%) of patients with joint pain was observed in patients over 60. The prevalence of joint pain in elderly patients, particularly in the age group 51–60 ($p = 0.021$) and above 60 years old ($p = 0.010$), was significantly higher than in the young age group (11–20 years; reference group).

## Comparison of real-time PCR, rapid IgM/IgG, and ELISA IgM/IgG for the diagnosis of CHIKV

In this study, real-time PCR and chikungunya IgM rapid tests were used to screen 1,806 suspected cases. CHIKV infection was confirmed in 1,295 samples by real-time RT-PCR, and/or rapid IgM assay was shown concerning the timing of sampling after the onset of disease (Fig 3). The samples positive by RT-PCR were higher than that of the rapid IgM assay if the sample was obtained during the first five days of disease onset. In contrast, the number of positive samples by using a rapid IgM assay was higher than that of RT-PCR if the sample was obtained after at least six days of disease. Our large sample size was advantageous, as we could perform

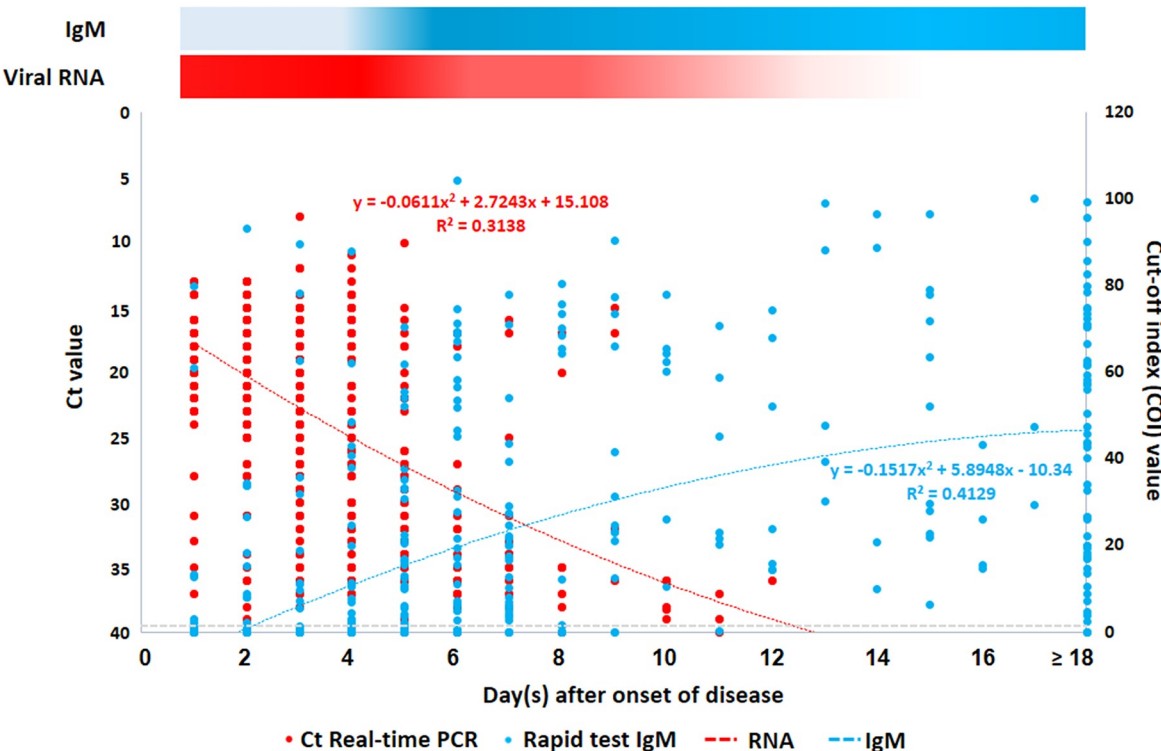

**Fig 3. Detection of CHIKV RNA by real-time RT-PCR and IgM antibody specific to CHIKV antigen by rapid test.** Results of 1,295 CHIKV infected patients were shown. Samples were tested by both real-time RT-PCR and rapid test for IgM antibody as described in the Materials and Methods. The Ct value and cut-off index (COI) represent the results of real-time RT-PCR and rapid IgM tests, respectively. COI is based on the ratio of assay signal to cut-off value of which COI≥1 = positive for anti-CHIKV IgM antibody and COI <1 = negative for anti-CHIKV IgM antibody. Curve equations and determination coefficients ($R^2$) are indicated.

**Table 2. Comparison between real-time PCR, rapid IgM/IgG, and ELISA IgM/IgG as detection methods for diagnosis of chikungunya.**

| Days after onset | Number | PCR + (%) | Rapid test | | ELISA | |
|---|---|---|---|---|---|---|
| | | | IgM+ (%) | IgG+ (%) | IgM+ (%) | IgG+ (%) |
| 1–3 | 148 | 139 (93.92) | 12 (8.11) | 5 (3.38) | 9 (6.08) | 4 (2.70) |
| 4–6 | 81 | 67 (82.72) | 30 (37.04) | 4 (4.94) | 31 (38.27) | 3 (3.70) |
| 7–9 | 34 | 12 (35.29) | 31 (91.18) | 4 (11.76) | 31 (91.18) | 8 (23.53) |
| 10–12 | 13 | 2 (15.38) | 13 (100.00) | 5 (38.46) | 13 (100.00) | 10 (76.92) |
| ≥13 | 34 | 1 (2.94) | 33 (97.06) | 22 (64.71) | 34 (100.00) | 31 (91.18) |
| **Total** | **310** | **221 (71.29)** | **119 (38.39)** | **40 (12.90)** | **118 (38.06)** | **56 (18.06)** |

real-time RT-PCR, rapid IgM/IgG, and ELISA IgM/IgG for CHIKV infection to demonstrate their sensitivity in disease onset. Serum samples from the first consecutive 310 patients confirmed to have CHIKV infection (RT-PCR positive = 221, and rapid IgM positive = 89) were selected to compare the sensitivity of real-time RT-PCR, rapid IgM/IgG, and ELISA IgM/IgG for CHIKV infection. The results of real-time RT-PCR, anti-CHIKV IgM/IgG by a rapid test and ELISA are presented according to days after onset in Table 2.

Among the 310 consecutive samples tested, 71.29%, 38.39%, and 38.06% were positive by RT-PCR, rapid IgM, and ELISA IgM, respectively. Although RT-PCR is considered a sensitive assay during the first few days after infection, 93.92% of the confirmed cases were positive at days 1–3. Rapid IgM and ELISA IgM were positive in 8.11% and 6.08% patients, respectively, during days 1–3. RT-PCR was positive in <40% patients from day 7 onwards following the onset of fever and was positive <10% from day 13 onwards following the onset of fever. Both IgM assays were positive in more than 90% of patients from day 7 onward following the onset of fever. Notably, an excellent agreement (Cohen's kappa value of 0.97) was observed between the rapid and ELISA IgM as demonstrated in S2 File. The percentages of real-time RT-PCR positive samples were <50% after 5 days, whereas the positive results by IgM antibody detection were >50% from day 6 after fever onset.

As shown in Table 2, the percentages of IgG-positive cases increased slower than those of IgM-positive cases. Among 310 consecutive samples tested, 12.90%and 18.06% were positive by rapid IgG and ELISA IgG, respectively. ELISA IgG was positive in 91.18% of patients after day 13, while rapid IgG was positive in 64.71% of patients after day 13. These results indicate that IgG detection by ELISA showed higher sensitivity than IgG detection by the rapid test. Nevertheless, the agreement between rapid and ELISA IgG showed Cohen's kappa value of 0.76, which represented the substantial agreement between rapid and ELISA IgG.

## Genotypic characterization of CHIKV detected in Thailand in 2018–2020

Phylogenetic tree analysis of CHIKV whole genomes was carried out from CHIKV isolated from six patients from the Central (Samut Sakhon, Bangkok, Samut Prakan); Eastern (Chonburi): Lower Northern (Tak); and Southern (Ranong) parts of Thailand with the reference CHIKV published previously in the GenBank database (Fig 4). Moreover, partial E1 phylogeny was also generated from 251 CHIKV isolated from various regions of the country, including the Central (Bangkok, Samut Sakhon, Samut Prakan, Nonthaburi); Eastern (Chonburi); and Southern (Phuket) regions of the country from October 2018 to February 2020 (Fig 5). From phylogenetic tree analysis, CHIKV isolates during the outbreaks in late 2018 to early 2020 belong to the Indian Ocean lineage within the ECSA genotype. However, these virus strains did not cluster with CHIKV strains previously circulating from the first massive outbreak in Thailand during 2008 to 2009. The current viruses were grouped with Bangladeshi strain of

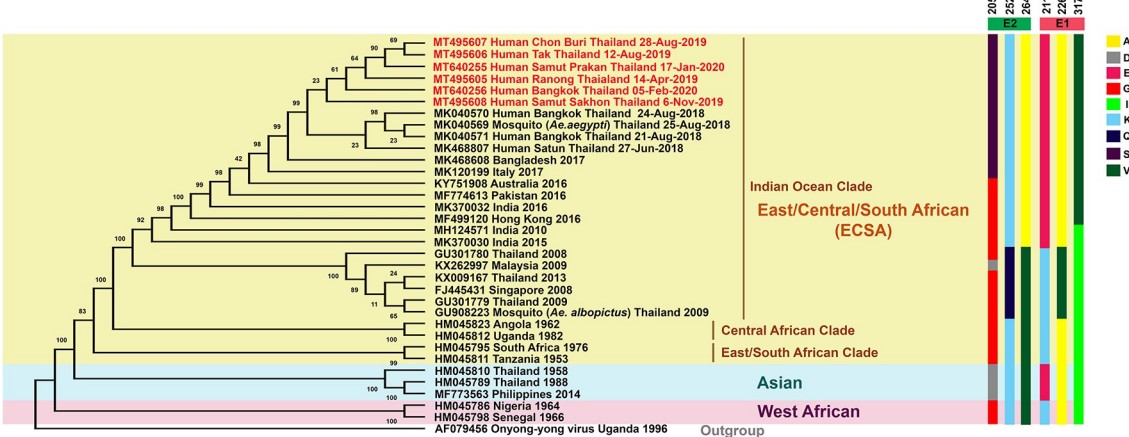

**Fig 4. Phylogenetic tree analysis of CHIKV whole genome.** Phylogenetic reconstruction of CHIKV whole-genome generated using the Maximum-Likelihood method with 1,000 ultrafast bootstrap replicates. Bootstrap values are represented at the branch nodes. The three major genotypes of CHIKV are highlighted in different colors. CHIKV strains isolated from this study are shown in red text (GenBank accession numbers MT495605-MT495608, MT640255, and MT640256). Bold lines in different colors show the specific amino acid substitutions in E2 and E1.

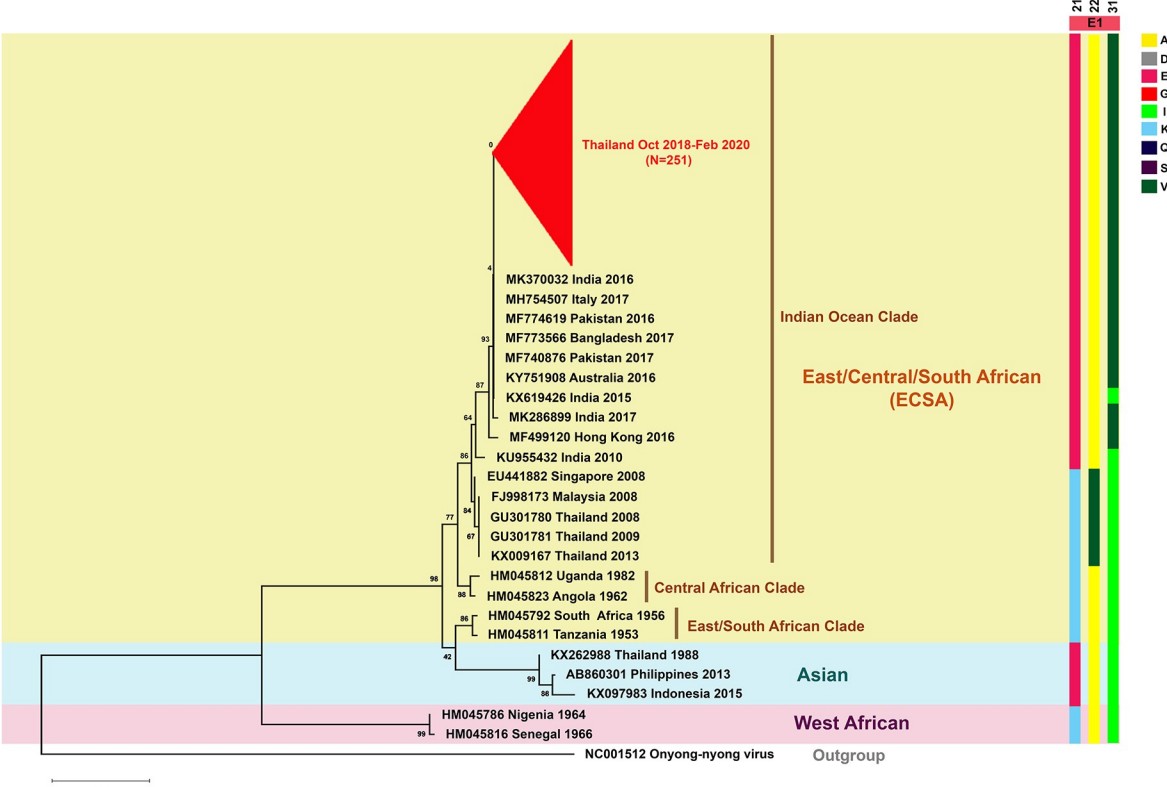

**Fig 5. Phylogenetic tree analysis of the partial CHIKV E1 gene.** A phylogenetic tree was constructed from partial E1 sequences of the 251 CHIKV isolates from this study and the selections from the GenBank database. The tree was constructed with MEGA X using the maximum-likelihood method with 1000 bootstraps. Bootstrap values are represented at branch nodes. The three major genotypes of CHIKV are highlighted in different colors. CHIKV isolated from this study is shown in red text. The specific amino acid substitutions in E1 are shown by bold lines in different colors.

2017, Italian strain of 2017, Indian strain of 2016, Pakistan strain of 2016, Hong Kong strain of 2016, and Australian strain of 2016 (Figs 4 and 5). We further investigated the similarity at nucleotide levels of the complete CHIKV genome among Thai strains in this present study (2018–2020) and the past outbreaks (1958, 1988, 2008, 2009, and 2013) as well as global ECSA strains (S3 File). The nucleotide similarity in each sequence of the present Thai strains showed 99.71%–99.96% similarity. In contrast, the ECSA Thai strains of 2008–2013 and the Thai strains of Asian genotype shared the range of percentage sequence similarity with the present Thai CHIKV strains as 98.91%–99.13% and 93.89%–94.72%, respectively. Moreover, the nucleotide similarity of the present Thai strains shared 98.95%–99.93% similarity with the sequence of global ECSA CHIKV strains. Interestingly, the complete genome of all CHIKV Thai strains showed a high identity with the Bangladeshi strain of 2017 (99.76–99.93%), especially the sequence of the Satun 2018 strain, which showed the most nucleotide sequence similarity with the Bangladeshi strain of 2017. The amino acid substitutions of the current Thailand strains were then analyzed and compared with the past ECSA Thai strains are shown in S4 File. In this study, we mainly focus on amino acid substitutions of CHIKV E1 and E2 proteins, which play crucial roles for virus entry into the host cell; moreover, change of amino acid in E1 and E2 is reportedly involved with vector adaptation. Interestingly, we found that all Thai strains of 2018–2020 carried alanine at the amino acid position 226 of the E1 gene. However, CHIKV isolated in Thailand in 2008, 2009, and 2013 showed alanine to valine substitution. Besides, CHIKV reported in this study carried the E1-K221E and E1-I317V mutations. The changes of amino acid residues in the E2 envelope protein, including G205S, K252Q, and V264A were observed. Additionally, the distinct amino acid mutations in the capsid and non-structural proteins (nsPs) between the present Thai strains and the past were located in the capsid (K73R), nsP2 (H130Y, E145D, N495S, S539L, and V793A), nsP3 (D372E), and nsP4 (S55N and R85G) (S4 File).

## Discussion

Although CHIKV was identified in 1952, there is still no specific antiviral drug or vaccine against CHIKV. CHIKV infection is not life-threatening, but infected patients commonly suffer from severe joint pain resulting in prolonged disability in some patients [18–20]. Arthralgia is a common clinical manifestation of patients hospitalized for CHIKV infection. Our study showed that older age is a significant risk factor for arthralgia associated with CHIKV infection. Accordingly, many studies have shown that increasing age is a predictive factor for the development of rheumatic sequelae in patients with chikungunya fever [21–23]. The use of proper laboratory tests for the diagnosis of CHIKV infection is essential for patient management and outbreak surveillance. We showed the usefulness of the detections of viral RNA and IgM specific to CHIKV antigen for diagnosis. Information on the duration of patient symptoms is crucial for the proper selection of laboratory testing. During the first week following infection, RT-PCR is the most appropriate test. However, patients who visit the doctor later after disease onset typically show negative RT-PCR results. The detection of the IgM antibody could be more useful in these cases. The IgG antibody is usually detectable later than the IgM antibody and is not commonly used for early diagnosis. The presence of IgG could be from past infection. Our study included a large group of patients whose samples were collected at different time points after disease onset. We concluded that for the diagnosis of chikungunya, qualitative real-time RT-PCR was highly sensitive between days 1–5 after the onset date, and IgM testing was highly sensitive after day 7. These findings corroborate with previous studies [24–26] as well as World Health Organization (WHO) guidelines for laboratory diagnosis of chikungunya [27]. Besides the laboratory diagnosis, the continuous surveys and studies of

genetic variation of CHIKV are crucial for infection control and understanding outbreaks for further development of the vaccine, specific antiviral drugs or prevention of future CHIKV outbreaks. Phylogenetic tree analysis revealed that CHIKV Thai strains isolated from patients in 2018–2020 belong to the ECSA genotype, the same genotype as CHIKV responsible for the massive outbreak in 2008–2009. However, genetic analysis of the present Thai strains of 2018–2020 showed marked differences compared to CHIKV reported in 2008–2009. CHIKV isolated from the massive outbreak in 2008–2009 in Thailand carrying E1-A226V mutation [28], whereas the causative agents for outbreaks before 2008 was caused by Asian genotype [17, 29]. We have previously reported that the CHIKV isolated from the massive outbreak in 2008–2009 clustered evolutionarily with the CHIKV strain of Sri Lanka, Singapore, and Malaysia [12]. Several studies have shown that E1-A226V mutation became more adapted for the CHIKV transmission by *Aedes albopictus* [30–32]. However, this variant would not sufficiently affect the evolution of the virus transmitted by *Aedes aegypti*. The mutation of E1-A226V enhances virus infectivity in the midgut cells of *Aedes albopictus*, resulting in increasing CHIKV dissemination and transmission by *Aedes albopictus* [33]. Remarkably, we found that none of the CHIKV ECSA strains isolated during the second massive outbreak in Thailand in late 2018 to early 2020 possess E1-A226V mutation. The present Thai strains of 2018–2020 harboring E1-226A without valine substitution at position 226 of E1 envelope glycoprotein were similarities to the first reported Asian genotype outbreak in 1958 in Thailand [11].

Interestingly, our study revealed that all present Thai strains of 2018–2020 possess two mutations, E1-K211E and E2-V264A, in conjunction with E1-226A. Our previous study also showed that CHIKV isolated from *Aedes aegypti* mosquito in Bangkok in August 2018 harboring these two mutations [15]. One document showing that E1: K211E variant was observed earlier in the CHIKV Asian genotype isolated from India in 1963 [34]. Double mutant virus containing E1-K211E and E2-V264A mutations in the background of E1-226A in the ECSA genotype was first observed in India in the year 2010 [35]. This mutated virus was also found in India in 2015–2017, Pakistan in 2016, Italy in 2017, and Bangladesh in 2017 [36–39]. There have been reported that positive selection had a dramatic effect on the alteration of amino acid residue from lysine (K) to glutamic acid (E) at position 221 of CHIKV E1 protein (E1: K211E) [40]. Previous studies reported that mutations on the E1 and E2 of CHIKV envelope glycoproteins impacted on vector competence, transmission efficiency, and virus pathogenicity [30, 32, 41–43]. Agarwal et al. demonstrated that combined E1-K211E and E2-V264A in the background of E1-226A was responsible for enhancing virus infection in *Aedes aegypti*, which led to more efficient CHIKV dissemination and facilitated the transmission of CHIKV in *Aedes aegypti* compared to wildtype CHIKV E1-226A. Nevertheless, these adaptive mutations in the background of E1:226A showed no significant impact on CHIKV fitness for *Aedes albopictus* mosquito [44]. Our study characterized the CHIKV isolated from various parts of Thailand, mostly from the patients in urban areas, where *Aedes aegypti* could be the most likely responsible vector. Therefore, present Thai strains harboring E1: K211E and E2: V264A in the background of E1:226A, responsible for the higher fitness for the *Aedes aegypti* vector [44], may facilitate viral circulation and persistence in the urban area of Thailand.

Notably, our study reveals that CHIKV Thai strains of 2018–2020 harboring E2-Q252K, while this mutation was not found in CHIKV responsible for the large scale outbreak in Thailand during 2008–2009. CHIKV Thai strains of 2008–2009 possessed K252Q substitution, associated with *Aedes albopictus* fitness [45, 46]. These results indicated that the present Thai strains might not harbor one of the *Aedes albopictus*-adaptive mutations. Moreover, substitutions in other amino acid residues such as E1-I317V and E2-G205S were observed in our study (S4 File). The E2-G205S detected in our isolates was also reported in the Italian isolates from the outbreak in Italy in 2017 [38]. Notably, the E1-I317V mutation, which was recently

found in India, Bangladesh, and Italy [38, 47–49], was detected in our study. Nevertheless, the significant impact of E1-I317V, E2-G205S, and E2-Q252K mutations on CHIKV evolution is still not known. Therefore, further investigation is needed to evaluate the role of these mutations.

A comparison of phylogenetic relationship of 6 whole-genome sequences and 251 partial E1 from CHIKV isolated during the 2018–2020 outbreaks in Thailand with other sequences available in the GenBank showed that all the sequences clustered with representative South Asian isolates of 2016–2017 rather than the CHIKV Thailand isolated from the previous outbreak in 2008–2009, indicating that the recent outbreak did not originate from the circulating strain in Thailand. Interestingly, the Bangladeshi strain of 2017 shared the highest nucleotide identity with the present Thai strains of 2018–2020. Overall, our results suggested that the recent outbreak of CHIKV in Thailand during 2018–2020 belongs to the ECSA genotype originating from other countries in South Asia, mostly Bangladesh.

## Conclusions

This study investigated a large population of CHIKV-infected patients to demonstrate their clinical profiles and the diagnostic approaches of CHIKV infection. We found that joint pain in chikungunya was widespread in infected patients after middle age and showed an increase in older patients. Regarding laboratory diagnoses of CHIKV infection, real-time PCR detection of CHIKV RNA was more sensitive for samples obtained within five days of disease onset, whereas IgM assay would be more sensitive for samples obtained after six days of fever onset. Moreover, our genetic analysis revealed the changes of CHIKV from the first massive outbreak during 2008–2009 in the country. The present CHIKV Thai strains are closely related to the viral strains from South Asia, especially Bangladesh. Here, our findings revealed that the second massive outbreak of CHIKV virus in Thailand occurred due to the importation of the viral strain from Bangladesh. The continuous surveys and studies of genetic variation of CHIKV are crucial for infection control and further development of specific drugs or prevention of CHIKV outbreaks.

## Supporting information

**S1 File. GenBank accession numbers.**
(DOCX)

**S2 File. Positive cases detected by real-time PCR and rapid IgM/IgG and ELISA IgM/IgG according to the days after onset.**
(PDF)

**S3 File. Percentage of nucleotide identity of CHIKV.**
(TIF)

**S4 File. Amino acid changes in CHIKV Thailand strain in the 2018–2020 outbreak compared to the Thailand strain of 2008–2013.**
(DOCX)

## Acknowledgments

We thank all staff for their technical and administrative assistance.

## Author Contributions

**Conceptualization:** Sarawut Khongwichit, Nasamon Wanlapakorn, Chintana Chirathaworn, Yong Poovorawan.

**Methodology:** Jira Chansaenroj, Thanunrat Thongmee.

**Supervision:** Yong Poovorawan.

**Writing – original draft:** Sarawut Khongwichit, Saovanee Benjamanukul.

**Writing – review & editing:** Chintana Chirathaworn, Yong Poovorawan.

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
