## [Decision Letter · Decision Letter 0]

2 Oct 2020

PONE-D-20-24139

Large-scale outbreak of Chikungunya virus infection in Thailand, 2018–2019

PLOS ONE

Dear Dr. Poovorawan,

Thank you for submitting your manuscript to PLoS One. Your submission has now been peer reviewed. I agree that the manuscript  would benefit from being revised according to the suggestions following and encourage you to do so. 

Editor Comments to the Authors:

Please see the reviewer's comments. 

We look forward to receiving your revised manuscript.

Kind regards,

José Moreira, MD, MSc

Academic Editor

PLOS ONE

Journal Requirements:

2. Please amend either the abstract on the online submission form (via Edit Submission) or the abstract in the manuscript so that they are identical.

4.We note that [Figure(s) 1] in your submission contain [map/satellite] images which may be copyrighted. All PLOS content is published under the Creative Commons Attribution License (CC BY 4.0), which means that the manuscript, images, and Supporting Information files will be freely available online, and any third party is permitted to access, download, copy, distribute, and use these materials in any way, even commercially, with proper attribution. For these reasons, we cannot publish previously copyrighted maps or satellite images created using proprietary data, such as Google software (Google Maps, Street View, and Earth). For more information, see our copyright guidelines: http://journals.plos.org/plosone/s/licenses-and-copyright.

1.    You may seek permission from the original copyright holder of Figure(s) [1] to publish the content specifically under the CC BY 4.0 license. 

Reviewers' comments:

Reviewer's Responses to Questions

**Comments to the Author**

1. Is the manuscript technically sound, and do the data support the conclusions?

Reviewer #1: Yes

2. Has the statistical analysis been performed appropriately and rigorously? 

Reviewer #1: Yes

3. Have the authors made all data underlying the findings in their manuscript fully available?

Reviewer #1: Yes

4. Is the manuscript presented in an intelligible fashion and written in standard English?

Reviewer #1: No

5. Review Comments to the Author

Reviewer #1: Review Comments to the Authors:

The manuscript: Large-scale outbreak of Chikungunya virus infection in Thailand, 2018–2019.

This is an interesting paper highlighting viral genetic and the evolution analysis of CHIKV in a location with consecutive expansions and reemergence of the disease, which was different from the strain of the last outbreak in the country during 2008-2009 but similar to the strain of the first outbreak in 1958 and the strain from Bangladesh strain of 2017.

Major strength includes a large number of patients collected from different parts of the country and a number of CHIKV E sequences analysis. The major weakness is that the article is not well written, the text is quite repetitive, and some of the references are not correctly referenced.

Introduction:

From line 87, this paragraph tries to mention the history of the CHIKV outbreak in Thailand. The mode of transmission should not be mentioned here (reference 11, line 89). Moreover, reference 11 did not report that the vector of CHIKV is Aedes aegypti (miss referenced??).

Line 48: The authors mention that the objectives of this study were to investigate the disease burden, emergent pattern, and genomic diversity. Word "disease burden" should be changed into a more appropriate word since the results reported in this paper were the number of cases during the outbreak (Figure 1 and figure 2) and clinical presentation (table 1, figure 3).

Materials and methods:

Materials and methods should be rewritten in a technical way, and some more details should be given, for example:

- Specimen collection: The previous history of CHIKV infection in the subject, to clarify if the presence of IgM or IgG was not the result of the past infection.

- Report on how the suspected cases were defined?

- Figure 1A and in "specimen collection", the authors described the sites where samples were collected but not indicated the number of cases collected from each area. The authors should mention them.

- The methods (detection of anti-CHIKV IgM and IgG antibodies by rapid test and by ELISA, L140 -159) might be referenced (if any).

- Chi-square test was also used (table 1), so it should be stated in data analysis (Methods).

Results:

The majority of data in figure 1 and figure 2 are unnecessary, and no important data (for example, figure 1B, 2A, 2B), and these are also described in the text.

Table 1 and figure 3 are repetitive data. So, table 3 is unnecessary. Moreover, figure 3 is unclear, and the description in the text is also confused (line 235-237).

Supplement 3 (the percentage of nucleotide identity of CHIKV) and supplement 4 (amino acid changes in CHIKV Thailand strain) are nicely detailed, and it is the useful information that supports the discussion and conclusion so that it should be shared or indicated in the result.

Line 253, change six age group into eight age groups.

Line 51: The authors state that "all samples were tested for the presence of CHIKV RNA, IgG, and IgM using…." wherein "Materials and methods" and "Results" (table 2) it indicate that IgG was tested only in 310 cases. So the results or methods should be re-explained.

Line 227: Even the number of females was higher than males, there is no significant difference between sex groups;

Line 214: change 20018-20019 to 2018-2019

Please recheck the number of days after onset (line 279); it is not coherent with the days in table 2.

Line 285: change 12.90% to 12.94%

Discussion:

Paragraph 4,5,6 of the discussion, the authors tried to discuss the genetic variation of CHIKV. Still, It isn't very clear and deserved to be shortened to be more focused on the meaning of the results compared to the literature. And some of the references need to be rechecked (especially reference 36, 37, 39, line 447).

Line 394: Sri Lanka or Pakistan? (recheck from the references)

Where is reference 28 in the text?

Line 442: indicate the location of the information (supplement 4)

How about the E2-G205S, E2-Q252K mutation in the background of E2, and E1-V226A, E1-1317V in the location of E1? It's unclear since the discussion (line 455) and supplement 4 seem to be non-coherent.

6. PLOS authors have the option to publish the peer review history of their article (what does this mean?). If published, this will include your full peer review and any attached files.

Reviewer #1: No

---

## [Author Response · Author response to Decision Letter 0]

22 Oct 2020

Journal Requirements:

This has been done

2. Please amend either the abstract on the online submission form (via Edit Submission) or the abstract in the manuscript so that they are identical.

This has been done

Ethics statement only appeared in the Methods section.

4.We note that [Figure(s) 1] in your submission contain [map/satellite] images which may be copyrighted. All PLOS content is published under the Creative Commons Attribution License (CC BY 4.0), which means that the manuscript, images, and Supporting Information files will be freely available online, and any third party is permitted to access, download, copy, distribute, and use these materials in any way, even commercially, with proper attribution. For these reasons, we cannot publish previously copyrighted maps or satellite images created using proprietary data, such as Google software (Google Maps, Street View, and Earth). For more information, see our copyright guidelines: http://journals.plos.org/plosone/s/licenses-and-copyright.

The original map published under the Creative Commons Attribution License (CC BY 4.0) (as shown in the link https://commons.wikimedia.org/wiki/File:Northern_Thailand_map_01.svg) was used in the revised manuscript.

Review Comments to the Author

Reviewer#1: Review Comments to the Authors:

The manuscript: Large-scale outbreak of Chikungunya virus infection in Thailand, 2018–2019.

This is an interesting paper highlighting viral genetic and the evolution analysis of CHIKV in a location with consecutive expansions and reemergence of the disease, which was different from the strain of the last outbreak in the country during 2008-2009 but similar to the strain of the first outbreak in 1958 and the strain from Bangladesh strain of 2017.

Major strength includes a large number of patients collected from different parts of the country and a number of CHIKV E sequences analysis. The major weakness is that the article is not well written, the text is quite repetitive, and some of the references are not correctly referenced.

Introduction:

From line 87, this paragraph tries to mention the history of the CHIKV outbreak in Thailand. The mode of transmission should not be mentioned here (reference 11, line 89). Moreover, reference 11 did not report that the vector of CHIKV is Aedes aegypti (miss referenced??).

The mode of transmission was removed, as the reviewer suggested. Reference # 11 was revised. 

11. Hammon WM, Rudnick A, Sather GE. Viruses associated with epidemic hemorrhagic fevers of the Philippines and Thailand. Science. 1960;131:1102–3

Line 48: The authors mention that the objectives of this study were to investigate the disease burden, emergent pattern, and genomic diversity. Word "disease burden" should be changed into a more appropriate word since the results reported in this paper were the number of cases during the outbreak (Figure 1 and figure 2) and clinical presentation (table 1, figure 3).

The word "disease burden" was removed, and this part was changed as shown below:

“Here, the clinical presentations in chikungunya, emergent pattern, and genomic diversity of the chikungunya virus (CHIKV) causing this massive outbreak were demonstrated.”

Materials and methods:

Materials and methods should be rewritten in a technical way, and some more details should be given, for example: 

Specimen collection: The previous history of CHIKV infection in the subject, to clarify if the presence of IgM or IgG was not the result of the past infection.

Currently, the laboratory diagnosis of CHIKV infection includes the detection of CHIKV RNA by real-time RT-PCR and IgM, IgG antibodies by the immunoassays. The presence of CHIKV RNA and/or CHIKV-specific IgM antibodies indicated CHIKV infection. The detection of CHIKV-specific IgG without IgM in suspected cases suggested that these patients had past CHIKV infection. This information was added to the revised manuscript. 

Report on how the suspected cases were defined?

Definition of the suspected cases was added in the revised manuscript. 

A suspected case of CHIKV was defined as a patient presenting with acute onset of fever (> 38.5°C) with or without severe joint pain and skin rash, particularly in a person who was residing, traveling, and working in an epidemic area with a high risk of CHIKV transmission.

Figure 1A and in "specimen collection", the authors described the sites where samples were collected but not indicated the number of cases collected from each area. The authors should mention them. 

As recommendation, we indicated the number of cases collected from each area in sample collection (specimen collection) and figure legend of figure 1:

Sample collection (specimen collection) as text “samples were collected from 13 provinces throughout the Central (Bangkok 1,137 cases, Samut Prakan 173 cases, Samut Sakhon 344 cases, Nonthaburi 1 case); Western (Ratchaburi 1 case, Kanchanaburi 1 case); Northern (Tak 1 case); Northeastern (Maha Sarakham 1 case); Eastern (Chon Buri 34 cases); and Southern (Ranong 1 case, Phuket 108 cases, Songkhla 1 case, Surat Thani 3 cases) regions of Thailand.” 

Figure legend of figure 1 as text “Fig 1. Map of Thailand showing the sample collection site in the 13 provinces. Map of Thailand showing sampling areas in Bangkok (1,137 cases), Samut Prakan (173 cases), Samut Sakhon (344 cases), Nonthaburi (1 case), Ratchaburi (1 case), Kanchanaburi (1 case), Tak (1 case), Maha Sarakham (1 case), Chon Buri (34 cases), Ranong (1 case), Phuket (108 cases), Songkhla (1 case), and Surat Thani (3 cases).” 

The methods (detection of anti-CHIKV IgM and IgG antibodies by rapid test and by ELISA, L140 -159) might be referenced (if any).

The reference for the ELISA method was included as reference 17:

17. Mendoza EJ, Robinson A, Dimitrova K, Mueller N, Holloway K, Makowski K, et al. Combining anti-IgM and IgG immunoassays for comprehensive chikungunya virus diagnostic testing. Zoonoses Public Health. 2019;66(8):909-17. doi: 10.1111/zph.12641.

Chi-square test was also used (table 1), so it should be stated in data analysis (Methods).

The chi-square test was stated in the data analysis, as the reviewer suggested. 

“A chi-square test and logistic regression analysis were performed to examine the association between patient demographics, including sex, age, and joint pain.”.

Results:

The majority of data in figure 1 and figure 2 are unnecessary, and no important data (for example, figure 1B, 2A, 2B), and these are also described in the text.

Figures 1B, 2A, and 2B were removed, as the reviewer suggested as the figures were unnecessary and described in the text.

Table 1 and figure 3 are repetitive data. So, table 3 is unnecessary. Moreover, figure 3 is unclear, and the description in the text is also confused (line 235-237).

-According to the repetitive data in Table 1 and Figure 3, figure 3 was removed. Moreover, the data from the table were used to describe in the text line 235-237. (Note: figure 4, 5, and 6 were changed to figure 3, 4, and 5, respectively) 

The previous description in the text is “As shown in Fig 3, the percentage of patients with joint pain increased with increasing age, of which more than 80% joint pain was found in patients >60 years of age, while <70% and 50% joint pain were found in patients aged 11–20 years and <10 years, respectively.”

The revised description is “As shown in Table 1, the presence of joint pain in CHIKV-confirmed cases was associated with age (p < 0.001). The highest percentage (81.5%) of patients with joint pain was observed in patients over 60. The prevalence of joint pain in elderly patients, particularly in the age group 51-60 (p = 0.021) and above 60 years old (p = 0.010), was significantly higher than in the young age group (11-20 years; reference group).” 

Supplement 3 (the percentage of nucleotide identity of CHIKV) and supplement 4 (amino acid changes in CHIKV Thailand strain) are nicely detailed, and it is the useful information that supports the discussion and conclusion so that it should be shared or indicated in the result.

-For supplement 3 (the percentage of nucleotide identity of CHIKV), we already share and indicate the percentage of nucleotide identity of CHIKV. 

- Amino acid changes in CHIKV Thailand strain (supplement 4) were indicated in the result as the reviewer suggested. 

“Additionally, the distinct amino acid mutations in the capsid and non-structural proteins (nsPs) of the present Thailand strains of 2018-2020 and Thailand strains of 2008-2013 were summarized in S4 Supporting Information. Of which, unique amino acid substitutions were located in the capsid (K73R), nsP2 (H130Y, E145D, N495S, S539L, and V793A), nsP3 (D372E), and nsP4 (S55N and R85G).”

The E1 and E2 mutations of present Thailand strains of 2018-2020 compared with Thailand strains of 2008-2013 were already indicated in the result.

Line 253, change six age group into eight age groups.

-It should be eight groups, as the reviewer mentioned. However, as the reviewer suggested that the data in Table 1 and Figure 3 are repetitive, figure 3 and the text in line 253 used to explain figure 3 were removed.

 Line 51: The authors state that "all samples were tested for the presence of CHIKV RNA, IgG, and IgM using…." wherein "Materials and methods" and "Results" (table 2) it indicate that IgG was tested only in 310 cases. So the results or methods should be re-explained.

Line 51: text “all sample” to “samples.” 

As the reviewer suggestion:

-In this study, real-time PCR and IgM/IgG rapid tests were performed to examine CHIKV viral RNA, anti-CHIKV IgM/IgG antibodies, respectively, in all sample. However, only serum from the first consecutive 310 CHIKV-infected patients whose sera samples were positive for CHIKV RNA by real-time PCR or positive for anti-CHIKV IgM by the rapid test were selected to determine anti-CHIKV IgM/IgG antibodies by using ELISA kit. 

-In method (Detection of anti-CHIKV IgM and IgG antibodies by the rapid test), we explained that “In this study, IgM/IgG antibodies against CHIKV in all serum samples were assessed using commercial fluorescence immunoassay on nitrocellulose membranes.” 

-In method (Detection of anti-CHIKV IgM and IgG antibodies by enzyme-linked immunosorbent assay (ELISA)), we explained that “In this study, the commercial ELISA kits (EUROIMMUN, Lu¨beck, Germany) [17] were performed to determine anti-CHIKV IgM/IgG antibodies with only serum from the first consecutive 310 CHIKV-infected patients whose sera samples were positive for CHIKV RNA by real-time PCR or positive for anti-CHIKV IgM by rapid test.”.

-As a result, we already explained that “Serum samples from the first consecutive 310 patients confirmed to have CHIKV infection (RT-PCR positive = 221, and rapid IgM positive = 89) was selected to compare the sensitivity of real-time PCR, rapid IgM/IgG, and ELISA IgM/IgG for CHIKV infection. The results of real-time PCR, anti-CHIKV IgM/IgG by a rapid test and ELISA are presented according to days after onset in Table 2.”.

Line 227: Even the number of females was higher than males, there is no significant difference between sex groups;

Statistical analysis was performed and mentioned in the revised manuscript. 

“According to this study, the percentages of CHIKV-confirmed cases was higher in females than males (57.8 vs. 42.2%). The presence of joint pain was slightly lower in males (70.2%) than in females (74.6%), but these differences did not reach statistical significance (p = 0.079 for sex and p = 0.089 for the presence of joint pain).”

Line 214: change 20018-20019 to 2018-2019

Figures 2A and 2B, including figure legends, were removed according to the reviewer's suggestion. 

Please recheck the number of days after onset (line 279); it is not coherent with the days in table 2.

-The number of days after onset was rechecked, and the text was revised as shown: “Real-time PCR was positive in <40% patients from day 7 onwards following the onset of fever and was positive <10% from day 13 onwards following the onset of fever.”

Line 285: change 12.90% to 12.94%

-We change 12.94% to 12.90% in the text, which coherent with the percentage in table 2

Discussion:

Paragraph 4,5,6 of the discussion, the authors tried to discuss the genetic variation of CHIKV. Still, It isn't very clear and deserved to be shortened to be more focused on the meaning of the results compared to the literature. And some of the references need to be rechecked (especially reference 36, 37, 39, line 447).

Paragraphs 4, 5, and 6 were rewritten, and the references were rechecked. 

Besides the laboratory diagnosis, the continuous surveys and studies of genetic variation of CHIKV are crucial for infection control and understanding outbreaks for further development of specific drugs or prevention of future CHIKV outbreaks. Evolutionary analysis revealed that CHIKV Thailand strains isolated from patients in 2018–2020 belong to the ECSA genotype, the same genotype as CHIKV responsible for the massive outbreak in 2008-2009. However, genetic analysis of CHIKV isolates during the outbreaks in late 2018 to early 2020 showed marked differences compared to CHIKV reported in 2008–2009. CHIKV isolated from the massive outbreak in 2008-2009 in Thailand belongs to the ECSA genotype carrying an alanine to valine substitution at E1 position 226 (E1-A226V) [25], whereas the causative agents for outbreaks before 2008 was caused by CHIKV belonging to the Asian genotype [18, 26]. We have previously reported that the CHIKV isolated from the massive outbreak in 2008-2009 clustered evolutionally with the CHIKV strain of Sri Lanka, Singapore, and Malaysia [12]. Several studies have shown that E1-A226V mutation became more adapted for the CHIKV transmission by Aedes albopictus [27-29]. However, this variant would not sufficiently affect the evolution of the virus transmitted by Aedes aegypti. The mutation of E1-A226V enhances virus infectivity in the midgut cells of Aedes albopictus, resulting in increasing CHIKV dissemination and transmission by Aedes albopictus [30]. Remarkably, we found that none of the CHIKV ECSA strains isolated during the second massive outbreak in Thailand in late 2018 to early 2020 possess E1-A226V mutation. The present Thailand strains of 2018-2020 harboring E1-226A without valine substitution at position 226 of E1 envelope glycoprotein were similarities to the first reported Asian genotype outbreak in 1958 in Thailand [11]. 

Interestingly, our study revealed that all present Thailand strains of 2018-2020 possess two mutations, E1-K211E and E2-V264A, in conjunction with E1-226A. Our previous study also showed that CHIKV isolated from Aedes aegypti mosquito in Bangkok in August 2018 harboring these two mutations [15]. One document showing that E1: K211E variant was observed earlier in the CHIKV Asian genotype isolated from India in 1963 [31]. Double mutant virus containing E1-K211E and E2-V264A mutations in the background of E1-226A in the ECSA genotype was first observed in India in the year 2010 [32]. This mutated virus was also found in India in 2015-2017, Pakistan in 2016, Italy in 2017, and Bangladesh in 2017 [33-36]. There have been reported that positive selection had a dramatic effect on the alteration of amino acid residue from lysine (K) to glutamic acid (E) at position 221 of CHIKV E1 protein (E1: K211E) [37]. Previous studies reported that mutations on the E1 and E2 of CHIKV envelope glycoproteins impacted on vector competence, transmission efficiency, and virus pathogenicity [27, 29, 38-40]. Agarwal et al. demonstrated that combined E1-K211E and E2-V264A in the background of E1-226A was responsible for enhancing virus infection in Aedes aegypti, which led to more efficient CHIKV dissemination and facilitated the transmission of CHIKV in Aedes aegypti compared to wildtype CHIKV E1-226A. Nevertheless, these adaptive mutations in the background of E1:226A showed no significant impact on CHIKV fitness for Aedes albopictus mosquito [41]. Our study characterized the CHIKV isolated from various parts of Thailand, mostly from the patients in urban areas, where Aedes aegypti could be the most likely responsible vector. Therefore, present Thailand strains harboring E1: K211E and E2: V264A in the background of E1:226A, responsible for the higher fitness for the Aedes aegypti vector [41], may facilitate viral circulation and persistence in the urban area of Thailand.

Notably, our study reveals that all present Thailand strains of 2018-2020 harboring E2-Q252K, while this mutation was not found in CHIKV responsible for the large scale outbreak in Thailand during 2008–2009. CHIKV Thailand strains of 2008-2009 possessed K252Q substitution, associated with Aedes albopictus fitness [42, 43]. These results indicated that the present Thailand strains of 2018-2020 might not harbor one of the Aedes albopictus-adaptive mutations. Moreover, substitutions in other amino acid residues such as E1-I317V and E2-G205S were observed in our study (S4 Supporting Information). The E2-G205S detected in our isolates was also reported in the Italian isolates from the outbreak in Italy in 2017 [35]. Notably, the E1-I317V mutation, which was recently found in India, Bangladesh, and Italy [35, 44-46], was detected in our study. Nevertheless, the significant impact of E1-I317V, E2-G205S, and E2-Q252K mutations on CHIKV evolution is still not known. Therefore, further investigation is needed to evaluate the role of these mutations. 

A comparison of phylogenetic relationship of 6 whole-genome sequences and 251 partial E1 from CHIKV isolated during the 2018–2020 outbreaks in Thailand with other sequences available in the GenBank showed that all the sequences clustered with representative South Asian isolates of 2016-2017 rather than the CHIKV Thailand isolated from the previous outbreak in 2008–2009, indicating that the recent outbreak did not originate from the circulating strain in Thailand. Interestingly, the Bangladesh strain of 2017 shared the highest nucleotide identity with all present Thailand strains of 2018–2020. Overall, our results suggested that the recent outbreak of CHIKV in Thailand during 2018–2020 belongs to the ECSA genotype originating from other countries in South Asia, mostly Bangladesh.

Line 394: Sri Lanka or Pakistan? (recheck from the references)

Where is reference 28 in the text?

-It should be Pakistan. However, we did revise the discussion in paragraphs 4, 5, and 6 so that the number of reference change to 34.

Line 442: indicate the location of the information (supplement 4)

-The location of supplement 4 was indicated as the reviewer mention. 

How about the E2-G205S, E2-Q252K mutation in the background of E2, and E1-V226A, E1-1317V in the location of E1? It's unclear since the discussion (line 455) and supplement 4 seem to be non-coherent.

-For E1-V226A is E1-226A, we already discuss the role of E1-226A together with two novel mutation E1-K211E and E2-V264A as the text :

“Agarwal et al. demonstrated that combined E1-K211E and E2-V264A in the background of E1-226A was responsible for enhancing virus infection in Aedes aegypti, which led to more efficient CHIKV dissemination and facilitated the transmission of CHIKV in Aedes aegypti compared to wildtype CHIKV E1-226A. Nevertheless, these adaptive mutations in the background of E1:226A showed no significant impact on CHIKV fitness for Aedes albopictus mosquito.” 

There have no report on the role of E2-G205S, E2-Q252K mutation in the background of E2, and E1- E1-1317V in the location of E1. So that in the discussion, we did describe the previous report that found E2-G205S, E2-Q252K, and E1- E1-1317V mutations before as the text below:

“Notably, our study reveals that all present Thailand strains of 2018-2020 harboring E2-Q252K, while this mutation was not found in CHIKV responsible for the large scale outbreak in Thailand during 2008–2009. CHIKV Thailand strains of 2008-2009 possessed K252Q substitution, associated with Aedes albopictus fitness [42, 43]. These results indicated that the present Thailand strains of 2018-2020 might not harbor one of the Aedes albopictus-adaptive mutations. Moreover, substitutions in other amino acid residues such as E1-I317V and E2-G205S were observed in our study (S4 Supporting Information). The E2-G205S detected in our isolates was also reported in the Italian isolates from the outbreak in Italy in 2017 [35]. Notably, the E1-I317V mutation, which was recently found in India, Bangladesh, and Italy [35, 44-46], was detected in our study. Nevertheless, the significant impact of E1-I317V, E2-G205S, and E2-Q252K mutations on CHIKV evolution is still not known. Therefore, further investigation is needed to evaluate the role of these mutations.”

---

## [Decision Letter · Decision Letter 1]

26 Jan 2021

PONE-D-20-24139R1

Large-scale outbreak of Chikungunya virus infection in Thailand, 2018–2019

PLOS ONE

Dear Dr. Poovorawan,

Thank you for submitting your manuscript to PLOS ONE. After careful consideration, we feel that it has merit but does not fully meet PLOS ONE’s publication criteria as it currently stands. Therefore, we invite you to submit a revised version of the manuscript that addresses the points raised during the review process.

Please follow the minor edition revisions indicated by the two reviewers. As Stated by reviewer 3, please smooth you statement line 358. Finaly, please answer to the reviewer about the phylogeny analysis and the choice of the reference sequences you did.

We look forward to receiving your revised manuscript.

Kind regards,

Pierre Roques, Ph.D.

Academic Editor

PLOS ONE

Reviewers' comments:

Reviewer's Responses to Questions

**Comments to the Author**

1. If the authors have adequately addressed your comments raised in a previous round of review and you feel that this manuscript is now acceptable for publication, you may indicate that here to bypass the “Comments to the Author” section, enter your conflict of interest statement in the “Confidential to Editor” section, and submit your "Accept" recommendation.

Reviewer #1: All comments have been addressed

Reviewer #2: All comments have been addressed

Reviewer #3: (No Response)

2. Is the manuscript technically sound, and do the data support the conclusions?

Reviewer #1: Yes

Reviewer #2: Yes

Reviewer #3: Yes

3. Has the statistical analysis been performed appropriately and rigorously? 

Reviewer #1: Yes

Reviewer #2: Yes

Reviewer #3: I Don't Know

4. Have the authors made all data underlying the findings in their manuscript fully available?

Reviewer #1: Yes

Reviewer #2: Yes

Reviewer #3: Yes

5. Is the manuscript presented in an intelligible fashion and written in standard English?

Reviewer #1: Yes

Reviewer #2: Yes

Reviewer #3: Yes

6. Review Comments to the Author

Reviewer #1: (No Response)

Reviewer #2: The study was comprehensive, well conducted and with thorough genomic analysis that highlights the massive outbreak in 2018-20 in Thailand due to imported strain from South Asia with mutations in E1 and E2 proteins, likely associated with vector adaptation which may facilitate more efficient CHIKV transmission. The authors have made significant revisions as advised by the reviewer.

My additional comments would be:

1.The manuscript can be much improved by removing redundancies in the text.

2. Suggest to include vaccines as well in Lines 387, 408.

3. Line 57 – the commercial immunoassay is the rapid test, suggest to mention as commercial immunoassay (rapid test).

4. Line 519 – suggest to change to “IgM assay either rapid test or ELISA” would be more sensitive…….

Reviewer #3: The article "Large-scale outbreak of Chikungunya virus infection in Thailand, 2018–2019" presents epidemiological surveillance data for the chikunungunya virus in Thailand over two years, using a large number of serum samples from patients with clinical symptoms compatible with this arbovirus. In addition to the molecular and serological diagnosis, the authors performed genomic surveillance and molecular characterization of possible mutations that led to changes in amino acids. The article was previously reviewed by other peer reviewers and the authors have already answered previous questions, as well as made the requested adjustments.

In view of what I read after the previous reviews, I consider the article to be very important, but I have some comments and questions.

Discussion

Lines 358 to 360: At the end of the first paragraph of the discussion, the authors write "We concluded that for the diagnosis of chikungunya, qualitative real-time RT-PCR was highly sensitive between days 1-5 after the onset date, and IgM testing was highly sensitive after day 7 ". This information, although it was verified during the research carried out by the authors, is not new. I request that the authors include that this information corroborates previous findings, even with the manual of the world health organization for the diagnosis of chikungunya.

Questions:

Throughout the results and discussion, the authors describe that the evolutionary analyzes were performed by maximum likelihood (ML), using the Mega X program and some available sequences from GenBank for the creation of the reference dataset.

- Why did you choose to use Mega X for an ML analysis knowing that it has limitations and demands a long time for the construction of the phylogenetic tree? Other programs and websites are available for building ML trees in a much faster and more robust way.

- What was the criterion for building the dataset with a reference sample? There are many more strings available on GenBank than are used to build the tree. Has any previous analysis been carried out with representative sequences from other regions that have had major epidemics in recent years? Why didn't they include it in this analysis?

- The phylogenetic tree figure shows the three main genotypes (ECSA, West African and Asia). In the two analyzes, both with the six sequences of the complete genome and with the 251 sequences of the E1 gene, the sequenced samples grouped with other sequences of the ECSA genotype, however in the Indian Ocean (IOL) lineage. One of the IOL markers is the E1-A226V mutation, which was not found in this study. Are the authors sure that their sequences belong to IOL, since other recent sequences of the ECSA genotype from other regions, in addition to the IOL lineage, were not included in the phylogenetic reconstruction?

- Why only sequenced E1 of other samples (in addition to six complete), since a focus of the study was the amino acid substitutions of E1 and E2 proteins?

- I suggest that in the future the authors do the phylogeographic and temporal reconstruction with a larger number of global sequences in order to demonstrate the place responsible for the introduction and to monitor and predict the route of dispersion in space and time of the strains circulating in Thailand.

7. PLOS authors have the option to publish the peer review history of their article (what does this mean?). If published, this will include your full peer review and any attached files.

Reviewer #1: No

Reviewer #2: No

Reviewer #3: No

---

## [Author Response · Author response to Decision Letter 1]

1 Feb 2021

Review Comments to the Author

Reviewer #1: (No Response)

Reviewer #2: The study was comprehensive, well conducted and with thorough genomic analysis that highlights the massive outbreak in 2018-20 in Thailand due to imported strain from South Asia with mutations in E1 and E2 proteins, likely associated with vector adaptation which may facilitate more efficient CHIKV transmission. The authors have made significant revisions as advised by the reviewer.

My additional comments would be:

1.The manuscript can be much improved by removing redundancies in the text.

As a suggestion, the redundancies in the text were removed as shown in the file of the revised manuscript with track changes.

2. Suggest to include vaccines as well in Lines 387, 408.

Thank you very much for the suggestion.

We included vaccines in the sentence below:

“Although CHIKV was identified in 1952, there is still no specific antiviral drug or vaccine against CHIKV.”

“Besides the laboratory diagnosis, the continuous surveys and studies of genetic variation of CHIKV are crucial for infection control and understanding outbreaks for further development of the vaccine, specific antiviral drugs or prevention of future CHIKV outbreaks.”

3. Line 57 – the commercial immunoassay is the rapid test, suggest to mention as commercial immunoassay (rapid test).

As the reviewer suggested, the word “commercial immunoassay” in the abstract was changed to commercial immunoassay (rapid test) as showed below:

“A total of 1,806 sera samples from suspected cases of chikungunya were collected from 13 provinces in Thailand, and samples were tested for the presence of CHIKV RNA, IgG, and IgM using real-time PCR, enzyme-linked immunoassay (ELISA), commercial immunoassay (rapid test).”

4. Line 519 – suggest to change to “IgM assay either rapid test or ELISA” would be more sensitive…….

As the reviewer suggested, the word “rapid test IgM” in the conclusions was changed IgM assay as showed below:

“whereas IgM assay would be more sensitive for samples obtained after six days of fever onset.”

Reviewer #3: The article "Large-scale outbreak of Chikungunya virus infection in Thailand, 2018–2019" presents epidemiological surveillance data for the chikungunya virus in Thailand over two years, using a large number of serum samples from patients with clinical symptoms compatible with this arbovirus. In addition to the molecular and serological diagnosis, the authors performed genomic surveillance and molecular characterization of possible mutations that led to changes in amino acids. The article was previously reviewed by other peer reviewers and the authors have already answered previous questions, as well as made the requested adjustments.

In view of what I read after the previous reviews, I consider the article to be very important, but I have some comments and questions.

Discussion

Lines 358 to 360: At the end of the first paragraph of the discussion, the authors write "We concluded that for the diagnosis of chikungunya, qualitative real-time RT-PCR was highly sensitive between days 1-5 after the onset date, and IgM testing was highly sensitive after day 7 ". This information, although it was verified during the research carried out by the authors, is not new. I request that the authors include that this information corroborates previous findings, even with the manual of the world health organization for the diagnosis of chikungunya.

Thank you very much for your suggestion.

We included the request information as the sentence below:

“These findings corroborate with previous studies and World Health Organization (WHO) guidelines for laboratory diagnosis of chikungunya.”

Questions:

Throughout the results and discussion, the authors describe that the evolutionary analyzes were performed by maximum likelihood (ML), using the Mega X program and some available sequences from GenBank for the creation of the reference dataset.

- Why did you choose to use Mega X for an ML analysis knowing that it has limitations and demands a long time for the construction of the phylogenetic tree? Other programs and websites are available for building ML trees in a much faster and more robust way.

We choose to use MEGA because it was performed to conduct evolutionary analysis in many studies in diverse biological fields. Moreover, this program easy to use with multiple features, including aligning sequence, test for selection model, building the phylogenetic tree, and estimating evolutionary distances.

- What was the criterion for building the dataset with a reference sample? There are many more strings available on GenBank than are used to build the tree. Has any previous analysis been carried out with representative sequences from other regions that have had major epidemics in recent years? Why didn't they include it in this analysis?

We would like to confirm the ECSA-IOL.

Most of the sequences that we selected belong to ECSA-IOL, responsible for the massive outbreak since 2006 in Asia and the virus's present strain that cause the recent outbreak in South and Southeast Asia. There is a very high probability that the virus circulating in South Asia and Southeast Asia is responsible for the current epidemic in Thailand than the virus in the other region such as Kenya (2016-2018), Republic of the Congo (2019). Moreover, the reference samples were also selected from the phylogenetic tree previously reported. Therefore, we know the cluster of reference samples used to build the dataset.

- The phylogenetic tree figure shows the three main genotypes (ECSA, West African and Asia). In the two analyzes, both with the six sequences of the complete genome and with the 251 sequences of the E1 gene, the sequenced samples grouped with other sequences of the ECSA genotype, however in the Indian Ocean (IOL) lineage. One of the IOL markers is the E1-A226V mutation, which was not found in this study. Are the authors sure that their sequences belong to IOL, since other recent sequences of the ECSA genotype from other regions, in addition to the IOL lineage, were not included in the phylogenetic reconstruction?

In this study, CHIKV belongs to ECSA-IOL lineage in phylogenetic reconstruction, including CHIKV harboring E1:226V (Singapore strain of 2008, Thailand of 2008, 2009 and 2013, Malaysia 2008 and 2009) and CHIKV possess two novel mutations in the background of E1:226A such as India 2016, Pakistan 2016, Bangladesh 2017 and Italy 2017). Our study showed that the present Thai strains were most closely related to the South Asian strain reported in 2016 and 2017 (India 2016, Pakistan 2016, and Bangladesh 2017) and Italy 2017, previously classified as ECSA-IOL. Therefore, we sure that the present Thai strains belong to the ECSA-IOL lineage. Additionally, other studies also confirmed that recent South Asian strain and Thai strain possess two novel mutations in the background of E1-226A, belong to the ECSA-IOL lineage (Intayot, Phumee et al. 2019, Lindh, Argentini et al. 2019, Eyase, Langat et al. 2020, Pyke, McMahon et al. 2020).

- Why only sequenced E1 of other samples (in addition to six complete), since a focus of the study was the amino acid substitutions of E1 and E2 proteins?

Because the previous studies showed that adaptive mutations in E1 envelope glycoproteins of CHIKV provide a fitness advantage in mosquito vector adaptation since the massive outbreak occurs in La Reunion. Moreover, our phylogenetic analysis showed a similar cluster of viral strains between the whole-genome and E1 gene. However, for future CHIKV surveillance, we plan to identify the sequence of both E1 and E2 as recent studies showed that dual mutations of E1 and E2 play a crucial role in vector adaptation.

- I suggest that in the future the authors do the phylogeographic and temporal reconstruction with a larger number of global sequences in order to demonstrate the place responsible for the introduction and to monitor and predict the route of dispersion in space and time of the strains circulating in Thailand.

Thank you for your kind suggestion.

---

## [Editor Report · Decision Letter 2]

5 Feb 2021

Large-scale outbreak of Chikungunya virus infection in Thailand, 2018–2019

PONE-D-20-24139R2

Dear Dr. Poovorawan,

We’re pleased to inform you that your manuscript has been judged scientifically suitable for publication and will be formally accepted for publication once it meets all outstanding technical requirements.

Kind regards,

Pierre Roques, Ph.D.

Academic Editor

PLOS ONE
---

## [Editor Report · Acceptance letter]

15 Feb 2021

PONE-D-20-24139R2 

Large-scale outbreak of Chikungunya virus infection in Thailand, 2018–2019 

Dear Dr. Poovorawan:

I'm pleased to inform you that your manuscript has been deemed suitable for publication in PLOS ONE. Congratulations! Your manuscript is now with our production department. 

Kind regards, 

on behalf of

Dr. Pierre Roques 

Academic Editor

PLOS ONE